# A Potential Anthelmintic Phytopharmacological Source of *Origanum vulgare* (L.) Essential Oil against Gastrointestinal Nematodes of Sheep

**DOI:** 10.3390/ani13010045

**Published:** 2022-12-22

**Authors:** Filip Štrbac, Slobodan Krnjajić, Maria Paola Maurelli, Dragica Stojanović, Nataša Simin, Dejan Orčić, Radomir Ratajac, Kosta Petrović, Goran Knežević, Giuseppe Cringoli, Laura Rinaldi, Antonio Bosco

**Affiliations:** 1Institute for Multidisciplinary Research, University of Belgrade, 11030 Belgrade, Serbia; 2Department of Veterinary Medicine and Animal Production, University of Naples Federico II, 80137 Naples, Italy; 3Department of Veterinary Medicine, Faculty of Agriculture, University of Novi Sad, 21102 Novi Sad, Serbia; 4Department of Chemistry, Biochemistry and Environmental Protection, Faculty of Sciences, University of Novi Sad, 21102 Novi Sad, Serbia; 5Scientific Veterinary Institute Novi Sad, 21113 Novi Sad, Serbia; 6Faculty of Applied Ecology, University of Metropolitan, 11030 Belgrade, Serbia

**Keywords:** gastrointestinal nematodes, anthelmintic resistance, essential oils, oregano, in vitro test, in vivo test

## Abstract

**Simple Summary:**

Inappropriate use of regular anthelmintic drugs has led to the development of anthelmintic resistance in sheep gastrointestinal nematodes (GINs) and consequently caused huge economic losses. Therefore, researchers worldwide are making efforts in finding novel strategies to control these parasites, which are mostly based on the rational use of synthetic drugs and the use of various alternatives, such as botanical anthelmintics. The aim of this study was to evaluate the anthelmintic efficacy of the essential oil (EO) of oregano (*Origanum vulgare* (L.)) against sheep GINs and to demonstrate the possibility of its use in veterinary practice. For obtaining clear results, both in vitro (egg hatch test) and in vivo efficacy tests (fecal egg count reduction test), as well as toxicity studies, on the hosts (clinical examination, blood count, and liver or kidney function test) were performed. The egg hatch test demonstrated the anthelmintic potential of the tested EO with an efficacy of 71.3%–93.7%, which was also demonstrated in the fecal egg count reduction test with an efficacy of 60.1% in total (on the second farm, 78.26%) shown on day 14 after treatment. Moreover, no negative or side effects of the applied EO formulation to the sheep were noticed in toxicity studies. The obtained results indicate the high potential of the *O. vulgare* EO for use in veterinary practice to control sheep GINs, as a part of an integrated strategy aimed to reduce the use of conventional anthelmintics. Therefore, these results may be significant for the future management of these infections.

**Abstract:**

The development of anthelmintic resistance in sheep gastrointestinal nematodes (GINs) requires novel strategies for the sustainable control of these parasites. This study aimed to evaluate the anthelmintic efficacy of the *Origanum vulgare* (L.) essential oil (EO) against sheep GINs and to evaluate the possibility of its use in control practice. The in vitro egg hatch test was conducted at eight different concentrations (50, 12.5, 3.125, 0.781, 0.195, 0.049, 0.025, and 0.0125 mg/mL) of the tested EO. For the in vivo fecal egg count reduction test, the EO of *O. vulgare* was administrated orally at a mean single dose of 150 mg/kg to sheep from two farms in Southern Italy, whereby potential toxic effects to the hosts were also evaluated. In the egg hatch test, the inhibition of egg hatchability varied from 71.3% to 93.7%, depending on the concentration used. The high anthelmintic potential was confirmed in the fecal egg count reduction test with an average reduction of nematode eggs in feces of 43.2% and 60.1% on days 7 and 14 after treatment, respectively. In addition, no toxic effects were noticed during the clinical examination of sheep or by observing blood count and liver or kidney function test results. The obtained results suggest the strong activity of the *O. vulgare* EO against sheep GINs, probably due to a high percentage of carvacrol (76.21%), whereby it can be considered safe for sheep at the dose tested in vivo. Therefore, it is suitable for use in veterinary practice as a part of an integrated strategy for the control of sheep GINs.

## 1. Introduction

Gastrointestinal helminthiasis in ruminants is caused by trichostrongylid worms that affect the digestive tract of animals [1]. In sheep, their parasitism in some cases may cause diarrhea, anemia, protein loss, anorexia, and reduced immunity, which can seriously endanger animal health and welfare, leading in some cases to death [2,3]. Although the infection often occurs subclinically, it can still result in impaired weight gain and yield of meat and milk, as well as in reduced fertility, which in turn also leads to economic losses [4,5,6]. For these reasons, gastrointestinal nematodes (GINs) represent a major obstacle to sheep production [1] with an annual cost of hundreds of millions of euros across Europe [7]. The most important sheep GINs that cause serious problems belong to the genera *Haemonchus*, *Ostertagia* (*Teladorsagia*), *Trichostrongylus*, *Nematodirus*, and *Cooperia* [8]. In some cases, *Oesophagostomum* and *Bunostomum* could be added to this list [4]. Nowadays, these parasites are widely distributed throughout the world, depending on geographic and climate conditions. For example, *Haemonchus* spp. and *Cooperia* spp. are distributed in subtropical and tropical regions, and *Ostertagia* spp. and *Nematodirus* spp. in temperate regions, while *Trichostrongylus* spp. are well represented throughout the world [8]. In Northern Serbia [9] and Southern Italy [10], the prevalence of these parasites is also high.

Currently, as well as in the past decades, the bases of the management of the infections caused by sheep GINs are anthelmintic drugs, such as benzimidazoles, macrocyclic lactones, and imidazothiazoles [11]. However, their intensive and incorrect use, which mostly refers to overfrequent treatments, underdosing, and the continuous use of only one drug, has led to the development of anthelmintic resistance (AR) [2,12], whereby single and multiresistant strains are now widely distributed throughout the world. In Europe, the annual costs of AR have been estimated at €38 million with a tendency to grow in the future [13]. The declining effectiveness of commercial drugs requests the devising of novel strategies, which are usually based on the integrated approach of controlling nematodes in an attempt to slow down the development and spread of AR and reach the appropriate efficacy. That implies, on the one hand, the rational use of anthelmintics, highlighting the importance of the incorporation of refugia-based strategies, such as target treatments (TT) and target-selective treatments (TST) [13,14,15], and, on the other hand, the use of various alternatives. That refers to a genetic selection of resistant animals, pasture management, balanced nutrition, biological regulation (the use of nematophagous fungi, bacteria, or even other nematodes), vaccine production, and the use of botanical anthelmintics [1,16,17].

The use of plants and their products is considered particularly promising in the control of sheep GINs [6,18]. Namely, in many countries with traditional cultures, ethnomedicine and phytotherapy are still the primary treatment options for many parasitic diseases [19]. Within that context, essential oils (EOs) of many plant species and their bioactive compounds are already initially evaluated for their anthelmintic potential [17,20]. These plant products are defined as aromatic, concentrated, and complex mixtures of volatile nonpolar compounds extracted from plant material [3,21,22] and, along with antiparasitic, have also antibacterial, antifungal, antiviral, anti-inflammatory, antiseptic, antioxidant, anticancer, and many other medicinal properties [21,23]. For that reason, the popularity of EOs in veterinary medicine has increased rapidly in the past years, whereby they are increasingly used for the treatment and prevention of various diseases and for improving animal production [21,24].

Within that context, oregano (*Origanum vulgare* L.) represents a perennial plant from the Lamiaceae family, which is rich in EO. It is a wild-growing or cultivated plant species of the Mediterranean coast, North Africa, and West Asia [25]. Due to its chemical composition, which consists of phenolic compounds, such as carvacrol and thymol in high percentages, the EO of oregano has already shown a wide range of biological and pharmacological activities [26,27,28]. Keeping that in mind, many possible indications for the use of the oregano EO in veterinary medicine are suggested or already proven: use in animal feed (against various pathogens in the gastrointestinal tract or for improving production performances in poultry, pigs, and ruminants); control and therapy of various diseases, such as coccidiosis, candidiasis, and mastitis; use for egg disinfection; different ecological purposes; or use in aquaculture [29,30]. However, little is still known about the use of oregano as an anthelmintic in animals. The aim of this study is to evaluate the anthelmintic activity of the oregano EO against sheep GINs and to evaluate the possibility of its use in clinical practice.

## 2. Materials and Methods

### 2.1. Essential Oil and Chemical Analyses

The essential oil of *O. vulgare* was obtained from the following producer: Bio salas Farago, Orom, Serbia. Qualitative and semiquantitative chemical characterization of the essential oil was performed by GC–MS at the Department of Chemistry, Biochemistry, and Environmental Protection, Faculty of Sciences, University of Novi Sad, Serbia, using an Agilent Technologies 6890N gas chromatograph coupled with an Agilent Technologies 5975B electron ionization mass-selective detector. The exact parameters for the analysis are described in Knežević et al. [31]. The compounds were identified by comparison of mass spectra with data libraries (Wiley Registry of Mass Spectral Data, 7th ed., and NIST/EPA/NIH Mass Spectral Library 05) and confirmed by comparison of linear retention indices with literature data [32]. The relative amount of each component is expressed as the percentage of its peak area relative to the total peak area. A homologous series of n-alkanes (C8-C28), injected under the same conditions, was used as a standard for the determination of retention indices.

### 2.2. Experimental Animals

The animals used for these studies were mainly the Lacaune/Bagnolese mixed breed dairy sheep, homogeneous in age (2 years ± 0.5) and grazing season, with an average body weight (b.w.) of 50 kg ± 5. Two farms were used, located in Southern Italy, in the Campania region, where the prevalence of sheep GIN is high [10]. For each test, animals from examined farms were randomly chosen and thus contained a different level of worm burdens (from 5 to 3820 EPG). Sheep tested were free of any anthelmintic treatments at least 6 months before trial. These tests were performed at the Regional Centre for Monitoring of Parasitosis (CREMOPAR, Eboli, SA) in Italy. The animals were fed with pasture and feed (grained barley and corn), whereby during the trial, the diet did not change. Before the administration of the essential oil, the animals were fasted and remained fasted until 2 h after the treatment (standard practice of administration of an anthelmintic). The study was conducted according to the guidelines of the Declaration of Helsinki and approved by the Ethics Committee of the University of Naples (PG/2021/0130480, 16 December 2021).

### 2.3. Egg Hatch Test—In Vitro Test

The in vitro anthelmintic potential of the oregano EO was evaluated using the egg hatch test. The GIN eggs were obtained from sheep with a natural-mixed infection using the recovery technique described by Bosco et al. (2018) [10]. Fecal samples were collected directly from the rectal ampulla of animals, after which they were processed within 2 h of collection. First, fecal samples were homogenized and filtered under running water through four sieves of different mesh sizes (1 mm, 250 μm, 212 μm, and 38 μm) to separate the eggs from the feces. Next, the GIN eggs retained on the last sieve were washed and centrifuged for 3 min at 1500 RPM with distilled water, after which the supernatant was discarded. In the end, centrifugation was performed using a 40% sugar flotation solution to float the eggs, which were then isolated in new tubes, mixed with distilled water, and then centrifuged two more times to remove pellets. In this way, an aqueous solution with GIN eggs was obtained [33].

The egg hatch test was performed as described by Ferreira et al. [34] and Štrbac et al. [3], with some modifications. Twenty-four-well plates, containing aqueous solutions of approximately 150 eggs/well, were used for the experiment. Eight different concentrations of the *O. vulgare* EO (50, 12.5, 3.125, 0.781, 0.195, 0.049, 0.025, and 0.0125 mg/mL) were emulsified in Tween 80 (3%, *v*/*v*) and completed with distilled water to reach a final volume of 0.5 mL in each tested well, whereby decreased concentrations of EO were obtained by the dilution method. After incubation for 48 h at 27 °C, the numbers of eggs and first-stage larvas (L1) were counted under an inverted microscope, and the inhibition of egg hatchability was calculated for each concentration and controls. The positive control was thiabendazole at the two lowest concentrations used for EO (0.025 and 0.0125 mg/mL), and the negative control were 3% Tween 80, *v/v* in a quantity used for emulgation, and distilled water. The experiment was performed in three replicates, whereby obtained values were expressed as an arithmetic mean ± standard deviation.

### 2.4. Fecal Egg Count Reduction Test—In Vivo Trial

The in vivo anthelmintic efficacy of the oregano EO was evaluated using the fecal egg count reduction test, which was performed on the same two farms that were used for the egg hatch test (Salerno Province, Italy). The procedure was performed with some differences compared with our last research [3]. Natural-mixed infection of GINs in the tested animals was confirmed using the fecal egg count (FEC), after which, on each farm, 36 sheep (*n* = 72 in total) were divided into three groups (*n* = 12 per group) and perorally treated in a single dose as follows:

Group 1 (G1): 150 mg/kg of b.w. of *O. vulgare* EO;

Group 2 (G2): albendazole in a standard dose (3.8 mg/kg) recommended by the manufacturer (positive control);

Group 3 (G3), sunflower oil, 50 mL per animal (negative control).

To avoid the effects of pure EO on the mucosa of the gastrointestinal tract (GIT), EO formulation was obtained by dilution of the oregano EO in sunflower oil (1:4.5). For the same reason, as well as to avoid the inactivation of EO compounds in proximal parts of the GIT, the oil was applied directly to the rumen of animals with a tube. Individual fecal samples were collected rectally before treatment (D0) and 7 and 14 days after treatment (D7 and D14) and stored shortly thereafter at 4 °C. The fecal samples were analyzed individually using the Mini-FLOTAC technique as described by Cringoli et al. (2017) [35], with a detection limit of 5 eggs per gram (EPG) of feces, and using a sodium chloride flotation solution (specific gravity = 1.200). The results were expressed as EPG in each sample with the calculation of averages ± standard deviation within each group at different time points (D0, D7, D14) to evaluate the effect of the tested formulation on the reduction of the number of eggs in feces. The final results are presented as arithmetic means from two tested farms.

### 2.5. Coproculture

To identify the GIN genera, the same quantity of feces was collected from each sample to create a pool for each fecal culture group at D0, D7, and D14, following the protocol described by the Ministry of Agriculture, Fisheries, and Food [36]. Developed third-stage larvae (L3) were identified using the morphological keys proposed by van Wyk and Mayhew [37]. The identification and percentages of each nematode genera were conducted on 100 L3; if a sample had 100 or less L3 present, all larvae were identified. Thus, on the total number of larvae identified, it was possible to give the percentage of each GIN genus.

### 2.6. Toxicity Studies (Hematological Parameters, Kidney and Liver Function)

For the evaluation of the potentially toxic and side effects of EO formulation, a clinical examination of tested animals was conducted at D0, D7, and D14, with special emphasis on general condition, animal behavior, feeding, and defecation. Additionally, blood samples from the jugular vein were taken from tested animals at D0 and D14 into a vacuum tube containing EDTA and used for the evaluation of toxic effects on blood count, whereby the analyses were conducted shortly thereafter (2–4 h after taking samples). Blood samples on the same days were also taken for biochemical analyses, for the evaluation of the effects of the applied EO formulation on the kidneys (parameters urea and creatinine) and the liver (activity of the enzymes aspartate aminotransferase (AST) and gamma-glutamyl transferase (GGT)).

A clinical chemistry system (Dimension RXL, Dade Behring, Ventura, CA, USA) was used to quantify urea, creatinine, and enzymes AST and GGT in serum samples. The white blood cell count, red blood cell count, hemoglobin, hematocrit, mean corpuscular volume, mean corpuscular hemoglobin, mean corpuscular hemoglobin concentration, platelets, lymphocytes, monocytes, neutrophil granulocytes, eosinophils granulocytes, and basophils granulocytes were determined in blood samples using an electronic particle counter (MS9; Melet Schloesing Laboratoires, Osny, France).

### 2.7. Statistical Analyses

The inhibition of egg hatchability (IH) was calculated using the following formula [16,38]: IH = number of eggs/(number of eggs + number of L1 larvae) × 100 (%)(1)

One-way analysis of variance (ANOVA) with post hoc Tukey’s test was performed for the mutual comparison of values obtained for different concentrations of the tested EO as well as with controls, with a *p*-value threshold of 0.05. For the evaluation of the presence of a dose-dependent effect, as well as for the calculation of half-maximal inhibitory concentration (IC_50_), nonlinear regression/logarithmic distribution was applied [34].

The reduction in fecal egg counts (FECR) was calculated using the following formula [39]:FECR = (1 − (T_2_/T_1_ × C_1_/C_2_)) × 100 (%) (2)
whereby T_1_ is EPG before treatment (D0) in the oregano or albendazole treatment group, T_2_ is EPG after treatment (D7 or D14) in the oregano or albendazole treatment group, C_1_ is EPG before treatment in the sunflower oil treatment group (C-), and C_2_ is EPG after treatment in the sunflower oil treatment group (C-).

The obtained values were compared within one group for different days, as well as within the same day for different groups using the two-way ANOVA, followed by Tukey’s test with a *p*-value threshold of 0.05.

For the analysis of the results of hematological and biochemical blood analyses, a two-way ANOVA was used, which was performed separately for each subject parameter. For the comparison of values in the same group on D0 and D14 after treatment, post hoc Sidak’s test (*p* < 0.05) was used, while for the comparison of values obtained for different groups on the same day, post hoc Tukey’s (*p* < 0.05) was used.

## 3. Results

### 3.1. Chemical Composition

A total of 10 compounds were identified by GC–MS analyses (Table 1, Figure 1). Carvacrol, a terpenoid from the group of phenols, was the most abundant ingredient (76.21%), followed by the hydrocarbon monoterpenes p-cymene (12.57%) and γ-terpinene (2.63%) and the sesquiterpene β-caryophyllene (2.23%). The rest of the compounds were present in less than 2%.

### 3.2. Egg Hatch Test—In Vitro Test

The essential oil of *O. vulgare* showed very potent in vitro ovicidal activity with inhibition of egg hatchability from 71.3% to 93.3%, depending on the used concentration. It should be emphasized that the effect was higher than 70% for each tested concentration (Table 2). Moreover, for the three highest concentrations, the effect was >90%, which is a similar activity to that of thiabendazole (both tested concentrations), whereby for an EO concentration of 0.781 mg/mL, the effect was similar to the 0.0125 mg/mL of thiabendazole (*p* > 0.05). All tested concentrations of *O. vulgare* showed significantly higher effects than both negative controls (*p* < 0.05). The calculated IC_50_ value was 0.15 mg/mL, and the R^2^ value was 0.95, suggesting its dose-dependent effect.

### 3.3. Fecal Egg Count Reduction Test—In Vivo Trial

The in vitro anthelmintic potential of the *O. vulgare* EO was confirmed with an in vivo trial, with an average reduction of nematode eggs in animal feces of 43.21% and 60.13% on D7 and D14, respectively (Table 3). Moreover, the treatment with the EO formulation significantly reduced the number of EPGs on both time points after treatment compared with D0, and also compared with the sunflower oil on D14 (*p* < 0.05), while the effect of albendazole was still higher. The effect of the oregano EO was higher at farm 2 with efficacies of 56.55% and 78.26% compared with farm 1 with efficacies of 29.87% and 41.99% on D7 and D14, respectively (Figure 2). Importantly, in terms of the reduction percentage of EPGs, the applied EO more affected the animals with a higher number of EPGs (>800) in comparison with lowered worm burden animals.

### 3.4. Toxicity Studies (Hematological Parameters, Kidney and Liver Function)

No toxic or side effects were observed during the clinical observation of animals treated with the *O. vulgare* EO. Furthermore, values of all hematological parameters after treatment were similar to that before treatment (*p* > 0.05), suggesting the absence of a toxic effect on the blood count (Appendix A). In the end, biochemical analyses of blood showed that values of urea, creatinine, and the activity of AST were not changed significantly, and the activity of GGT was lower at D14 compared with D0, suggesting that the treatment with the EO did not affect the kidney and liver function (Table 4).

### 3.5. Coproculture

All the collected samples were qualitatively analyzed for the nematode’s composition. The genera of nematodes detected in two farms, in all groups (*O. vulgare* EO, albendazole, and sunflower oil), and in all days (D0, D7, and D14) of sampling were *Haemonchus, Trichostrongylus, Teladorsagia*, and *Chabertia*, as shown in Table 5 and Table 6.

## 4. Discussion

The demonstration of the anthelmintic efficacy of botanical preparations requires the use of reliable methods and tests. First, in vitro tests are important for the initial evaluation of the anthelmintic potential of some active substances and are good for the selection of the most appropriate candidates for further testing [40,41]. Since the EO of oregano was shown as the most effective in our last research [3], it was chosen for further analyses. In vitro tests are not intensive and expensive [34], given that they are much more often conducted in previous studies than in vivo tests [17,19,42]. However, field testing provides a more realistic picture of the efficacy of potential anthelmintics and their possibility for use in practice, thus necessary for the development of anthelmintic drugs. The choice of tests is also important, whereby in studies that include the evaluation of the anthelmintic activity of plant products, the egg hatch test as in vitro and the fecal egg count test as in vivo are recommended due to their accuracy and reliability [38,43]. Finally, toxicity studies are important to prove the safety of using a new preparation, since it must not have any negative effects on the hosts. In addition, GC–MS analyses are crucial for the examination of EO composition and the identification of compounds with anthelmintic activity. Therefore, all of these steps are important for performing before the implementation of an anthelmintic agent in practice to control GINs.

On the egg hatch test, the *O. vulgare* EO showed great ovicidal activity varied from 71.3 to 93.7%, whereby four different concentrations showed a similar effect to that of thiabendazole. Moreover, three concentrations showed an effect higher than 90%, which is an in vitro efficacy needed for a drug to be considered effective for the control of nematodes, including GINs, according to the criteria that were set up by the World Association for the Advancement of Veterinary Parasitology (WAAVP) [41,44]. Moreover, the obtained IC_50_ concentration of 0.15 mg/mL is one of the lowest obtained for some EO in such studies, suggesting its great potential. In our first study, the EO of oregano showed a maximal inhibitory effect (100%) on the egg hatch inhibition for concentrations ranging from 0.049 to 50 mg/mL [3]. In a study by Jimenez-Penago et al. [45], the EO of *O. vulgare* showed a similar effect against sheep GINs (*H. contortus*) as in the present study, with a concentration needed to reduce 50% hatching of 0.17–0.28 mg/mL, although the chemical composition of that sample was different (consists mainly from eugenol, 76.3%). In the same study, the EO also showed high activity against GINs isolated from cattle (*Cooperia* spp.).

The first thing that can be noticed by observing results in the fecal egg count reduction test are the high values of the standard deviation of EPGs, which suggest the big differences in worm burdens in animals. That can be explained by the fact that in small ruminants, most of the parasite population (approximately 80%) are aggregated and dispersed in only 20–30% of hosts, while the majority of animals have low worm burdens [5,46]. Anyhow, the anthelmintic potential of the *O. vulgare* EO shown in the egg hatch test was confirmed in the fecal egg count reduction test with a total reduction of the number of nematode eggs in animal feces of 43.21% and 60.13% on D7 and D14 after treatment with only a single dose, respectively. In Figure 2, it can be noticed that the effect was better on farm 2, where the FECR reached 56.55% and 78.26% on D7 and D14 after treatment. These differences can be explained by the different ways of keeping animals, which was free range on farm 1, while on farm 2, sheep were in boxes, which facilitated the manipulation of animals and the application of the EO formulation. Other variables, such as feed or volume of rumen content, may also have been involved.

It may be also noticed that the shown in vitro efficacy of the *O. vulgare* EO was higher than that shown in vivo. However, this finding is often a case in similar studies [44,47] and can be explained by many factors that affect the efficacy of EOs in field conditions. That includes the anatomical and physiological specifics in the ruminant gastrointestinal tract [48], as well as the instability of EOs (metabolic transformations in the gut) [21,49]. Both of these may lead to the inactivation of EO active ingredients to some extent before they reach the target in the abomasum and especially in the intestine [3]. This state may be confirmed by the fact that many studies so far demonstrated that EOs showed the greatest activity against *H. contortus* in comparison with GINs that live in more distal parts of the gastrointestinal tract [50,51,52]. However, this problem may be overcome with the use of the encapsulation technique to improve the efficacy and stability of EOs [46,50] or by increasing the dose or prolonging the therapy for several days, especially when the tested EO did not show any toxic or side effects, such as the case in the present study. Additional studies should be conducted on the effects of the oregano EO on each particular GIN genus to prove its effectiveness against various nematodes and to confirm the above-mentioned states.

The effects of the oregano EO showed both that in vivo and in vivo may be considered superior in comparison with EOs from other plants so far tested for anthelmintic efficacy [17,20], where in some cases, oils did not show any in vivo activity or reach some efficacy at higher doses. In comparison with our last study, some progress with oregano has been made given that the EO of thyme and the binary combination of linalool/estragole showed in vivo efficacy of around 25% at a dose of 100 mg/kg [3]. The biological effects of oregano are most likely related to a high percentage of carvacrol, since the anthelmintic effects of EOs are usually associated with a major component [40,53]. Indeed, the activity of isolated carvacrol against sheep GINs is already demonstrated in different studies so far. In a study by Andre et al. [51], isolated carvacrol showed high ovicidal and larvicidal activity against *H. contortus* with IC_50_ values of 0.17 and 0.2 mg/mL, respectively. Furthermore, in the same study, its acetylate derivate, carvacrol acetate, showed also high in vivo activity with a reduction of EPG of 35.4% and 65.9% on days 8 and 16 after treatment, respectively, against a mix of different GIN species (*Haemonchus* spp. (90%), *Trichostrongylus* spp. (7%), and *Oesophagostomum* spp. (3%)). The superior activity of carvacrol was also confirmed in a study by Katiki et al. [54], where it exhibited one of the strongest ovicidal activities against *H. contortus* in comparison with other tested compounds, with an IC_50_ value of 0.11 mg/mL.

The exact act of nematocidal activity of carvacrol is still not fully understood, although it can ensure useful information on the most appropriate formulation [55]. Within this perspective, several different mechanisms of action of carvacrol are suggested in a study by Andre et al. [51], such as the changes in the cuticle of parasites that may affect the permeability of the cuticle and motility, structural changes in the external reproductive organs of female worms that may affect nematode reproduction and be responsible for reducing the production of eggs, and the effect on pH value in abomasum that favors transticular absorption by parasites. Moreover, carvacrol may cause neurotoxic effects on the nematodes, since it interacts with SER-2 tyramine receptors, as demonstrated against the free-living nematode *Caenorhabditis elegans* in a study by Lei et al. [56]. In studies by Marijanović et al. [57,58], it was demonstrated that carvacrol exhibits antagonist properties of the nicotinic acetylcholine receptor (nAChR) of *Ascaris suum* and significantly inhibits the contractions of the neuromuscular preparation, but also potentiates the inhibitory effects of GABA. In the same studies, it was also demonstrated that carvacrol has synergistic effects with piperazine and monepantel on the contraction of the neuromuscular preparation of *A. suum*. These findings suggest that carvacrol, and therefore also the oregano EO, may find their use in a combination of drug agonists of nAChR, such as imidazothiazoles, or drugs agonists of GABA receptors, such as avermectins and piperazine.

The clinical evaluation of tested animals in the present study, as well as conducted hematological and biochemical analyses, suggests the absence of toxic and side effects of *O. vulgare* on animal behavior, as well as kidney and liver function, in the dose applied in vivo (150 mg/kg). However, it must be noted that the results of the present toxicity study are still only initial and must be supplemented in further studies with necropsy and gross and histopathological data. Long-term effects of potential oregano application in practice, if it will be used, should also be followed. Although very important for the evaluation of the possibility of the use of new potential anthelmintic agents, toxicity studies of botanical preparations were rarely conducted especially on the sheep hosts. Therefore, little is known so far about the effects of oregano on hosts, but the results of the present study preliminary suggest its safety. In rats, for example, it was shown that the EO of *O. vulgare* may cause signs of reproductive toxicity by interfering in sperm and affecting hormonal parameters, causing infertility in male Wistar rats but only at very high doses, while moderate doses were claimed safe [59]. Moreover, these concentrations also did not affect the physical, motor, and neuroendocrine parameters of rat dams and their offspring [60].

Acute toxicity of orally administrated carvacrol and carvacrol acetate in mice was evaluated in a study by Andre et al. [51], with obtained LC_50_ values of 919 and 1544.5 mg/kg, respectively. That suggests, for example, lowered toxicity of acetylated derivates of EO compounds, as also shown for thymol [40]. Similar toxicity studies as in the present study on sheep were conducted by Katiki et al. [47,61], whereby authors showed that the EO of lemongrass (*Cymbopogon schoenanthus*) is safe for sheep at doses of 180 and 360 mg/kg, as well as an encapsulated combination of anethole and carvone at doses of 20 and 50 mg/kg for lambs, since no significant differences among group mean for the hepatic (enzymes) or kidney (urea and creatinine) parameters were recorded after treatment with the EO, or the value was lower (GGT). A similar trend is noticed in the case of hematological parameters and animal behavior. However, one animal treated with a lower dose of lemongrass showed signs of liver toxicity, although it cannot be surely related to the application of the EO. Nevertheless, all mentioned findings suggest that toxicity studies should be implemented in every study that attempts to develop a new anthelmintic agent, even if it is plants or their product.

As discussed earlier, the high anthelmintic efficacy of the *O. vulgare* EO originates from the phenolic compound carvacrol, but also the hydrocarbon terpenes p-cymene, γ-terpinene, and β-caryophyllene and their synergism. A wider number of compounds in EOs in general with the different possible mechanisms of action may provide a high activity against various parasite stages. At the same time, this may lead to their less susceptibility to resistance in comparison with commercial drugs [34], mainly consisting of only one active substance. Another advantage of botanical anthelmintics may be lower toxicity for hosts and the environment, and especially in the aspect of leaving fewer residues in meat and milk [34,62], although these states are controversial and need confirmation in further studies. Finally, the sustainability of the use only of commercial drugs is also questionable from the financial aspects, since the prices of drugs continue to rise [63,64]. Additionally, from that perspective, a wide number of medicinal plants and their products offer an opportunity to find the most appropriate one from a financial point of view, especially in countries with developed biodiversity [19,34]. Therefore, although additional studies are required, the efficacy achieved in the present study (>90% in vitro and >60% in vivo) with no side effects on animal behavior, as well as liver and kidney function, suggests that the oregano EO may be the new, additional anthelmintic means used in the practice against sheep GINs.

However, still little is known about the use of EOs in veterinary medicine for various purposes, including anthelmintic [19]. For this reason, there is a steady increase in the number of such studies in general [65]. Additionally, although in some cases there were problems with in vivo efficacy of applied EOs, this problem may be solved with an encapsulation technique or a possible different way of use, as mentioned above. Moreover, the simultaneous use of EOs with commercial drugs may lead to their synergism [57,58], which increases the efficacy of the drugs. On the other hand, that can be important from the aspect of reducing the use of commercial drugs and slowing down the development and spread of anthelmintic resistance [6]. In the end, an integrated approach for controlling parasites may involve two various alternatives, as demonstrated in a study by Keeton [66], where the plant sericea lespedeza as a supplement in feeding was successfully combined with copper oxide wire particles applied as TST for reducing the number of *Coccidia* in ewes and lambs. Since, in the present study, the EO of *O. vulgare* mostly reduced the number of nematode eggs in animals with high EPGs, it may also be applied as TST in combination with some other alternative in an attempt to control sheep GINs. Therefore, the use of EOs and other plant products along with the rational use of anthelmintics or other alternatives offers various options for sustainable control of nematodes including GINs in future treatments.

## 5. Conclusions

The development and wide spread of anthelmintic resistance require devising new strategies in the management of nematode infections. Within that context, the use of botanical anthelmintics, such as EOs, represents a promising alternative. In the present study, the EO of *O. vulgare* showed great in vitro but also in vivo anthelmintic activity against sheep GINs, while, at the same time, it did not show any toxic or side effects to the hosts at the dose tested. The great anthelmintic potential of *O. vulgare* is derived from the high percentage of carvacrol, but also other compounds and their synergism. Therefore, preliminary results suggest that it can be used in practice as a part of the integrated control of sheep GINs, which will provide sustainable management of these infections by reducing the use of commercial anthelmintics and slowing down the spread of resistance. Further studies should be based on the development of an encapsulated pharmaceutical form of the formulation, which may further enhance in vivo efficacy, and thus possibly be used even as an independent product.

## Figures and Tables

**Figure 1 animals-13-00045-f001:**
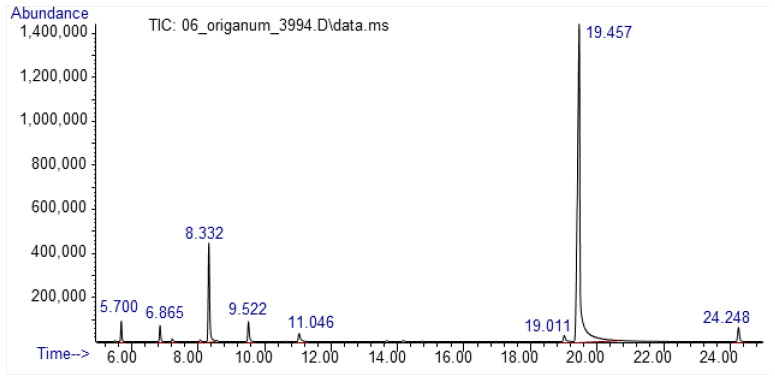
GC–MS chromatogram of the *Origanum vulgare* essential oil.

**Figure 2 animals-13-00045-f002:**
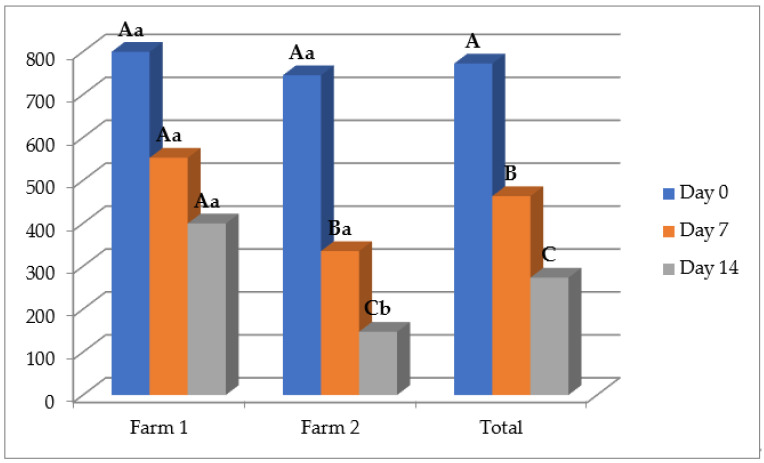
Mean number of eggs per gram at different time points after the treatment in a group of animals treated with the *Origanum vulgare* essential oil; uppercase compares means between different time points on the same farms, and lowercase between different farms on the same time points. Different letters indicate significant differences (*p* < 0.05).

**Table 1 animals-13-00045-t001:** Chemical composition (% of total peak area) of the *Origanum vulgare* L. essential oil determined by GC–MS.

AI *	Compound	% of Total Peak Area
932	α-Pinene	1.81
976	β-Pinene	1.64
990	Myrcene	0.32
1016	α-Terpinene	0.20
1023	p-Cymene	12.57
1057	γ-Terpinene	2.63
1100	Linalool	1.43
1292	Thymol	0.97
1303	Carvacrol	76.21
1418	β-Caryophyllene	2.23
Total % of identified compounds		100

* AI—arithmetic retention index.

**Table 2 animals-13-00045-t002:** The inhibitory effect (mean ± standard deviation) of different concentrations of the *Origanum vulgare* essential oil on egg hatching of sheep gastrointestinal nematodes.

Concentration of EO (mg/mL)	Inhibition of Hatchability (%)
50	93.3 ± 2.52 ^AB^*
12,5	93.7 ± 1.53 ^AB^
3.125	91.3 ± 1.53 ^AB^
0.781	89.0 ± 1 ^BC^
0.195	84.0 ± 1 ^CD^
0.049	80.0 ± 1 ^DE^
0.025	77.7 ± 1.53 ^E^
0.0125	71.3 ± 1.53 ^F^
Control (+) ^a^	96.3 ± 1.53 ^A^
Control (+) ^b^	95.0 ± 1 ^AB^
Control (−) ^c^	14.2 ± 3.34 ^G^
Control (−) ^d^	6.6 ± 1.92 ^H^

* Uppercase compares means between different concentrations and controls. Different letters indicate significant differences (*p* < 0.05). Control (+) ^a^—thiabendazole, 0.025 mg/mL; control (+) ^b—^thiabendazole, 0.0125 mg/mL; control (−) ^c^—3% Tween 80, *v*/*v*; control (−) ^d^—distilled water; EO—essential oil.

**Table 3 animals-13-00045-t003:** EPG (mean ± standard deviation) of sheep gastrointestinal nematodes and efficacy (reduction %) of the *Origanum vulgare* essential oil based on fecal egg count reduction test—in total from both examined farms.

Treatment		Day 0	Day 7	Day 14
*O. vulgare*, 150 mg/kg	EPG	772.3 ± 865.8 ^Aa^*	463.9 ± 405.9 ^Ab^	273.4 ± 265.0 ^Ac^
	Reduction	/	43.21%	60.13%
Albendazole, 3.8 mg/kg (control +)	EPG	683.1 ± 735.5 ^Aa^	9.38 ± 11.8 ^Bb^	36.89 ± 41.6 ^Bc^
	Reduction	/	98.98%	95.06%
Sunflower oil, 50 mL (control −)	EPG	914.0 ± 821.9 ^Aa^	914.4 ± 784.8 ^Aa^	800.5 ± 680.3 ^Cb^
	Reduction	/	/	/

* Uppercase compares means between different groups at one time point; lowercase compares means of different time points within one group. Different letters indicate significant differences (*p* < 0.05); EPG—eggs per gram.

**Table 4 animals-13-00045-t004:** Impact of the *Origanum vulgare* essential oil on the parameters that indicate kidney and liver function in tested animals—in total from both examined farms.

Group	Day	Urea (mg/dL)	Creatinine (mg/dL)	AST (UI/L)	GGT (UI/L)
*Origanum vulgare* EO	0	31.0 ± 7.41 ^Aa^*	13.58 ± 3.68 ^Aa^	164.4 ± 48.64 ^Aa^	72.75 ± 7.62 ^Aa^
	14	28.42 ± 6.71 ^Aa^	14.08 ± 3,53 ^Aa^	173.1 ± 46.08 ^Aa^	66.42 ± 4.72 ^Ba^
Albendazole (control +)	0	27.58 ± 9.45 ^Aa^	12.42 ± 2.75 ^Aa^	173.3 ± 82.72 ^Aa^	72.00 ± 8.86 ^Aa^
	14	28.42 ± 6.07 ^Aa^	13.42 ± 3.68 ^Aa^	163.0 ± 50.72 ^Aa^	66.50 ± 5.54 ^Ba^
Sunflower oil (control −)	0	29.83 ± 7.87 ^Aa^	12.42 ± 2.84 ^Aa^	189.7 ± 72.63 ^Aa^	72.50 ± 7.82 ^Aa^
	14	29.33 ± 4.91 ^Aa^	12.83 ± 3.04 ^Aa^	206.9 ± 56.32 ^Aa^	73.67 ± 5.30 ^Ab^

* Uppercase compares means between values at different time points within one treatment group and lowercase values at the same time points between different treatment groups. Different letters indicate significant differences (*p* < 0.05); AST—aspartate aminotransferase, GGT—gamma-glutamyl transferase; EO—essential oil.

**Table 5 animals-13-00045-t005:** Percentage of sheep nematode third-stage larvae (L3) for each group at D0, D7 and D14 in sheep farm 1.

Group	Day	*Haemonchus* (%)	*Trichostrongylus* (%)	*Teladorsagia* (%)	*Chabertia* (%)
*Origanum vulgare* EO	D0	10	53	32	5
	D7	08	44	39	9
	D14	07	35	48	10
Albendazole (control +)	D0	13	39	42	6
	D7	-	-	-	-
	D14	-	-	-	-
Sunflower oil (control −)	D0	12	36	45	7
	D7	09	34	47	10
	D14	10	31	48	11

**Table 6 animals-13-00045-t006:** Percentage of sheep nematode third-stage larvae (L3) for each group at D0, D7, D14, and D21 in sheep farm 2.

Group	Day	*Haemonchus* (%)	*Trichostrongylus* (%)	*Teladorsagia* (%)	*Chabertia* (%)
*Origanum vulgare* EO	D0	11	42	44	3
	D7	05	47	43	5
	D14	06	43	45	6
Albendazole (control +)	D0	11	44	43	2
	D7	-	-	-	-
	D14	-	-	-	-
Sunflower oil (control −)	D0	13	44	41	2
	D7	11	40	45	4
	D14	08	42	47	3

## Data Availability

All data generated or analyzed during this study are included in this published article. The datasets used and/or analyzed during the present study are available from the corresponding author upon reasonable request.

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
