# Peer review of "A Potential Anthelmintic Phytopharmacological Source of *Origanum vulgare* (L.) Essential Oil against Gastrointestinal Nematodes of Sheep"

_animals, 2022, doi:10.3390/ani13010045_

Round 1

Reviewer 1 Report

The search for alternatives to control parasites in animals is very important to prevent the advancement of parasitic resistance to anthelmintics. I congratulate the authors about the research topic.Here are some suggestions for improving the text.

Materials and Methods

Line 185: Why did you use sunflower oil as a negative control?

Line 266: Do the percentages of reduction in the in vivo test, presented in table 3, correspond to the average of the two farms? Please make this clearer in the text.

Author Response

  1. Line 185: Why did you use sunflower oil as a negative control?

Sunflower oil was used as a negative control given that it is used for dilution of essential oil before in vivo application. The quantity of sunflower oil corresponds with the quantity of formulation (oregano EO + sunflower oil in 1:4.5) given to the animals (50 ml/animal).

  1. Line 266: Do the percentages of reduction in the in vivo test, presented in table 3, correspond to the average of the two farms? Please make this clearer in the text.

Yes, now we emphasized it in Table 3 and also in Table 4. On Figure 1, the results are presented from both examined farms separately and in total.

Reviewer 2 Report

In this manuscript, several concerns need to be addressed as follows:

1.      Keywords: replace EHT; FECRT with their full term.

2.      Lines 182: on what basis the authors have chosen the tested dose of f O. vulgare EO.

3.      Line 215-216: How have the liver and kidney function indicators been measured? Illustrate the methods used or the full data of the kits used.

4. Adding the oil chromatogram in the result section is highly recommended.

5.      Page 7: table 1 has been repeated again.

6.      Figure 1: the presentation of the figure is very confusing and not reflecting any statistical analysis. Also, the full term of EPG should be clarified and the unit of measurement.

7.      Line 285: values of all hematological parameters after treatment 285 were similar to that before treatment (p>0.05). Where are the results of the hematological parameters??? No tables or figures exist for these parameters.

8.      In all tables’ footnotes: The full term of all abbreviations used within the table should be clarified in the footnote.

Author Response

  1. Keywords: replace EHT; FECRT with their full term.

Now it is replaced.

  1. Lines 182: on what basis the authors have chosen the tested dose of O. vulgare EO?

We have carefully checked the in vivo studies to date and their results (efficacy percentage against nematodes, effects on the hosts and used dosage), whereby some of those studies are cited in this paper. Also, we have chosen the tested dose based on our experience from our last research, where we used a lower dose (100 mg/kg) and obtained some efficacy (around 25%) for different essential oils. Therefore, we wanted to test another oil as well to find a dose to reach an even higher efficacy, still with no side effects on the animals.

  1. Line 215-216: How have the liver and kidney function indicators been measured? Illustrate the methods used or the full data of the kits used.

Now it is added in the section Material and methods, subsection Toxicity studies (haematological parameters, kidney and liver function).

  1. Adding the oil chromatogram in the result section is highly recommended.

Now it is added.

  1. Page 7: table 1 has been repeated again.

Accidentaly, now it has been deleted.

  1. Figure 1: the presentation of the figure is very confusing and not reflecting any statistical analysis. Also, the full term of EPG should be clarified and the unit of measurement.

New Figure 1 with statistical analyses is added instead the old one. EPG means eggs per gram, which is actually one of the units of measurement of worm burdens in animals.

  1. Line 285: values of all hematological parameters after treatment 285 were similar to that before treatment (p>0.05). Where are the results of the hematological parameters??? No tables or figures exist for these parameters.

This table is now added to supplementary material. We avoided putting it into the manuscript given its size.

  1. In all tables’ footnotes: The full term of all abbreviations used within the table should be clarified in the footnote.

Full terms of abbreviations are added to the caption of the tables.

Reviewer 3 Report

The article is very interesting, try to look for new therapeutic solutions in combating the invasion of gastrointestinal nematodes in sheep.
The described scientific experiment is performed according to accepted standards.
 I only have a few comments on the editing of this article.
The authors in the introduction (lines 65-67) describe nematodes from the Trichostrongylidae family as gastrointestinal nematodes. In my opinion, there is a lack of emphasis on the distinctness of the genus Nematodirus, which belongs to the Molineidae family and is a separate, often more and more dominant problem in many countries.
 In the chapter "Material and Methods", the authors, in my opinion, described the methodology of testing essential oils in too much detail (verses 120-128) - this is not a biochemical work, but a parasitological one.
 In my opinion, this chapter should include a sub-section - "experimental animals", with a description of the specifics of the maintenance of the two herds used for the study, and information on the separation of groups of animals from the entire herd.
Please check the animal figures in the FECRT test. ( line 180 ) 3 x 12 is 36 , why the group consisted of 60 animals .
 It is incomprehensible why the oils were administered through a tube into the rumen. (verse 188). This is additional stress for the animals. No digestive processes take place in the esophagus. If anything, they should be fed into the abomasum.
(line 211) whether the authors obtained permission from the ethics committee to draw blood. (needle stick stress). Such experiments in the EU require the approval of the ethics committee.
In the discussion, please refer to the relatively small diversity of types of gastrointestinal nematodes. The genus Nematodirus has not been shown. and Oesophagostomum.
(verse 349) about what animal manipulations the author walks about, explaining the differences in results in two herds.

Author Response

The authors in the introduction (lines 65-67) describe nematodes from the Trichostrongylidae family as gastrointestinal nematodes. In my opinion, there is a lack of emphasis on the distinctness of the genus Nematodirus, which belongs to the Molineidae family and is a separate, often more and more dominant problem in many countries.

We agree that the genus Nematodirus is very important and deserves attention. However, our coproculture examination showed the presence of only four gastrointestinal nematode genera: Haemonchus, Trichostrongylus, Teladorsagia and Chabertia on both tested farms. Therefore, we focused on the GIN genera presented in our research.

In the chapter "Material and Methods", the authors, in my opinion, described the methodology of testing essential oils in too much detail (verses 120-128) - this is not a biochemical work, but a parasitological one

The primary focus of this work is actually on pharmacology (an attempt to find a novel anthelmintic therapeutic agent against sheep GINs) and parasitology. However, this is multidisciplinary research whereby the evaluation of the chemical composition of used essential oil is very important to see what active ingredients the oil consists of and which ones are responsible for anthelmintic action (in this case carvacrol). This part is indispensable in such studies attempt to find a new natural preparation. Nevertheless, we agree that the focus should be on pharmacology and parasitology and therefore the subsection „Essential oil and chemical analyses” is now shortened as much as possible.

In my opinion, this chapter should include a sub-section - "experimental animals", with a description of the specifics of the maintenance of the two herds used for the study, and information on the separation of groups of animals from the entire herd.

This subsection is now added and separated from other subsections.

Please check the animal figures in the FECRT test. ( line 180 ) 3 x 12 is 36 , why the group consisted of 60 animals

Thanks to the reviewer for noticing that mistake, now is fixed.

It is incomprehensible why the oils were administered through a tube into the rumen. (verse 188). This is additional stress for the animals. No digestive processes take place in the esophagus. If anything, they should be fed into the abomasum.

Compounds of essential oils are prone to degradation and inactivation in the digestive tract including proximal parts. Therefore, our idea is to avoid these processes in these parts to reach a higher efficacy and to ensure enough quantities of active ingredients in target places (the abomasum and intestines). For example, in our last research [3] we applied essential oils using a syringe direct to the mouth and we obtained in vivo efficacy of around 25% (although in that study we used a lower dose, 100 mg/kg). There is also a risk of shedding and spitting oils applied in this way. We think that this way of application may overcome these problems although we are aware it is not ideal either. Therefore, along with the choice of oils and the dose, finding the best way to their application is also important, and our idea is to try different ways through different studies. Thanks to the reviewer for suggesting fed into the abomasum, we are also thinking about it for our next research. There is also an option for applying through the food, this may be least stressful for the animals.

Whether the authors obtained permission from the ethics committee to draw blood. (needle stick stress). Such experiments in the EU require the approval of the ethics committee.

We have permission from the Ethics Committee of the University of Naples (PG/2021/0130480 del 16/12/2021) to conduct research. This is stated in Institutional Review Board Statement at the end of the manuscript.

In the discussion, please refer to the relatively small diversity of types of gastrointestinal nematodes. The genus Nematodirus has not been shown. and Oesophagostomum.

As we mentioned earlier, those genera are important but in our study different genera were present (Haemonchus, Trichostrongylus, Teladorsagia and Chabertia).

(verse 349) about what animal manipulations the author walks about, explaining the differences in results in two herds.

We are talking about a different way of animal keeping. On Farm 1, animals were free range and therefore the application was harder than in Farm 2, where animals were kept in boxes that allow an easier application of oils. In the discussion section, there is a such explanation.

Reviewer 4 Report

Authors have presented a good study regarding use of phytochemical/natural source as anthelmintic in veterinary practice.

Authors are requested to address/change/edit following queries:

1) Please remove the 'toxicity study' from the title, because no such complete toxicity study has been carried out. the biochemical parameters can presented as in vivo study. toxicity study requires to follow OECD guidlines, parameters like necropsy, gross and histopathological data are very important to call it toxicity study. the present data (enzyme parameters) can be said as part of in vivo study  which is good and data is in support of the finding of this study. So remove from the title and other part of manuscript.

2) Introduction can be reorganized (may not be, it is upto the authors) by addressing (authors have already mentioned, but it can be if they want by addresing in as paragraph on..... ), to counter the..... emergance of resistance, alternative of medication, primary or first aid treatment, local use of medicinal plant, researching new treatment for future etc.....

2) please mention the animal ethics permission for conducting in vivo experiment in material methods,

3) Authors have done well by performing GC analysis.

4)result presented well.

5) Discussion part... need high attention, need to reduce in lengh and precise to the context of study. in vivo and in vitro study have given initial indication of the potential effect. so discussion should start by focusing the chemical compounds found by GC analysis, must address it,

Second, address in vitro and in vivo study, (as you have done already but with focusing on your and previous literature and conclude based on that)

Thirdly, write the limitations of the study, what needs to be done in future, and then try to infer the study with balance view because there is no proper toxicity study however, you can balance it by addressing the biochemical parameters (as enzymes are the primary indication of any abnormalities associate with organ).... the reduction was noted around 60% and there was no alteration in biochemical parameters, it can be presented as... the essential oil can be used as prelimnary/emergency/initial treatment (as you have very well mention in conclusion section). Avoid any interpretation which you haven't done in present study or it can be write in such a way that it should not be absolutely correct. 

So discussion and conclusion need major attention, reorganized and present well with addressing each parameters . Future study may be done by using particular components of essential oil and by targeting perticular nematode.

Please refer following article for your references and may incorporate :

1) https://www.researchgate.net/publication/323562505_Benzimidazole_Resistance_An_Overview

2) https://www.researchgate.net/publication/348280839_ANTHELMINTIC_DRUG-RESISTANT_DETECTION_METHODS_A_BRIEF_OVERVIEW

3) https://journals.plos.org/plosone/article?id=10.1371/journal.pone.0097053#s2

4) https://parasitesandvectors.biomedcentral.com/articles/10.1186/s13071-014-0518-2

5) https://www.sciencedirect.com/science/article/abs/pii/S0278691517300017?via%3Dihub

Author Response

1) Please remove the 'toxicity study' from the title, because no such complete toxicity study has been carried out. the biochemical parameters can presented as in vivo study. toxicity study requires to follow OECD guidlines, parameters like necropsy, gross and histopathological data are very important to call it toxicity study. the present data (enzyme parameters) can be said as part of in vivo study  which is good and data is in support of the finding of this study. So remove from the title and other part of manuscript.

We agree with the reviewer that we did not conduct a complete toxicity study. For example, long-term side effects also were not followed. Our focus this time was on short-term effects on liver and kidney function, which are mostly affected by some drugs, in addition to following haematological parameters. Therefore, we removed toxicity study from the title, but also in vitro and in vivo. We assume that is not necessary to emphasize in the title what kind of tests we did since there is a chapter Material and methods. Also, now we change the title of the subsection „Toxicity study“ in this chapter to „Toxicity studies (haematological parameters, kidney and liver function)“, to further clarify what we examined.

2) Introduction can be reorganized (may not be, it is upto the authors) by addressing (authors have already mentioned, but it can be if they want by addresing in as paragraph on..... ), to counter the..... emergance of resistance, alternative of medication, primary or first aid treatment, local use of medicinal plant, researching new treatment for future etc.....

In this section, we tried to emphasize the significance of gastrointestinal nematodes in sheep, the emergence of the problem of anthelmintic resistance, the potential alternatives etc. Special focus was on the essential oils (and the oregano especially) and their general possible importance in veterinary medicine... Although they are very well known for their medicinal properties, exact and scientific data on the use of the medicinal plant is still very scare.

3) please mention the animal ethics permission for conducting in vivo experiment in material methods,

Ethics permission is stated in Institutional Review Board Statement at the end of the manuscript, and now is also added to section Material and methods.

4) Authors have done well by performing GC analysis.

Thanks, we recognized the importance of determining the chemical composition in studies that refer to the finding of the novel plant-based anthelmintic therapeutic means.

5) result presented well.

Thanks. As other reviewer recomended, in new version we added a statistic into graph 1 and  essential oil chromatogram. The table with haematological parameters into suplementary material is also added.

6) Discussion part... need high attention, need to reduce in lengh and precise to the context of study. in vivo and in vitro study have given initial indication of the potential effect. so discussion should start by focusing the chemical compounds found by GC analysis, must address it,

Second, address in vitro and in vivo study, (as you have done already but with focusing on your and previous literature and conclude based on that)

Thirdly, write the limitations of the study, what needs to be done in future, and then try to infer the study with balance view because there is no proper toxicity study however, you can balance it by addressing the biochemical parameters (as enzymes are the primary indication of any abnormalities associate with organ).... the reduction was noted around 60% and there was no alteration in biochemical parameters, it can be presented as... the essential oil can be used as prelimnary/emergency/initial treatment (as you have very well mention in conclusion section). Avoid any interpretation which you haven't done in present study or it can be write in such a way that it should not be absolutely correct.

So discussion and conclusion need major attention, reorganized and present well with addressing each parameters . Future study may be done by using particular components of essential oil and by targeting perticular nematode.

Many thanks to the reviewer for his suggestions. In the new version, we tried to further improve the discussion section. Not major changes were done, but some parts are added, some were deleted and some are changed/moved. In this section, firstly, we tried to describe the procedure of discovering and testing the novel anthelmintic agent before its incorporation into the practice. After that, we individually described results from each test (in vitro test, in vivo test, part of toxicity tests conducted, chemical analyses and the coproculture) with the addition of some parts that are related to it (f.e. a possible act of nematicidal activity of tested EO). In each of this paragraph, we tried to discuss the results which include its strength (very high nematicidal activity especially in comparison with similar trials conducted so far), their limitation (f.e. efficacy reached in vivo could be further improved), the ways of improving that (use of encapsulation technique or multiple uses during few days) which also implies idea for further trials, and also the limitations of the study (uncomplete toxicity study). Where possible, we compared our results with the results so far obtained in similar research. In the end, we tried to summarize by stating the advantages and disadvantages of the use of EOs against sheep GINs, and what could also be done (f.e. the use of EOs with commercial drugs or with other alternatives to achieve sustainable control of GINs). In general, in this section, we tried to connect all the results obtained in one story and to explain why they can be important. Some of the literature sources recommended by the reviewer are also added.

Round 2

Reviewer 2 Report

No further comments to be addreesed 

Author Response

Many thanks to this and other reviewers for all of their suggestions. We believe that it further improved our manuscript. In the latest version, we corrected the rest of the suggestions and fixed some grammar mistakes we found. 

Reviewer 4 Report

Authors have incorporated almost all suggestions. Article can be accepted now.

Author Response

(The authors gave the same response as above.)
